

# JlBox v1.0: A Julia based mixed-phase atmospheric chemistry box-model

Langwen Huang[1,2] and David Topping[2]

[1]Department of Mathematics, ETH Zurich, Switzerland
[2]Department of Earth and Environmental Science,The University of Manchester, UK

**Correspondence:** Langwen Huang (langwen.huang@math.ethz.ch)

**Abstract.**

As our knowledge and understanding of atmospheric aerosol particle evolution and impact grows, designing community mechanistic models requires an ability to capture increasing chemical, physical and therefore numerical complexity. As the landscape of computing software and hardware evolves, it is important to profile the usefulness of emerging platforms in tackling this complexity. Julia is a relatively new programming language that promises computational performance close to that of Fortran, for example, without sacrificing flexibility offered by languages such as Python. With this in mind, in this paper we present and demonstrate the initial development of a high-performance community mixed phase atmospheric 0D box-model, JlBox, written in Julia.

In JlBox v1.0 we provide the option to simulate the chemical kinetics of a gas phase whilst also providing a fully coupled gas-particle model with dynamic partitioning to a fully moving sectional size distribution, in the first instance. JlBox is built around chemical mechanism files, using existing informatics software to provide parameters required for mixed phase simulations. In this study we use mechanisms from a subset and the complete Master Chemical Mechanism (MCM). Exploiting the ability to perform automatic differentiation of Jacobian matrices within Julia, we profile the use of sparse linear solvers and pre-conditioners, whilst also using a range of stiff solvers included within the expanding ODE solver suite the Julia environment provides, including the development of an adjoint model. Case studies range from a single volatile organic compound [VOC] with 305 equations to a 'full' complexity MCM mixed phase simulation with 47544 variables. Comparison with an existing mixed phase model shows significant improvements in performance and potential for developments in a number of areas.

## 1 Introduction

Mechanistic models of atmospheric aerosol particles are designed, primarily, as a facility for quantifying the impact of processes and chemical complexity on their physical and chemical evolution. Depending on how aligned these models are with the state-of-the-science, they have been used for validating or generating reduced complexity schemes for use in regional to global models (Zaveri et al., 2008; Riemer et al., 2009; Amundson et al., 2006; Korhonen et al., 2004; Roldin et al., 2014; Hallquist et al., 2009; Kokkola et al., 2018). This is based on the evaluation that 'full' complexity schemes are too computationally expensive for use in large scale models. With this in mind, the community has developed a spectrum of box-models that focus on





a particular process or experimental facility (e.g., Riemer and Ault, 2019), or use a combination of hybrid numerical methods
to capture process descriptions for use in regional to global models (e.g., Zaveri et al., 2008; Kokkola et al., 2018). Recent
studies are also exploring coupling the latter with numerical techniques for reducing systematic errors through assimilation of
ambient measurements (e.g., Sherwen et al., 2019).

With ongoing investments in atmospheric aerosol monitoring technologies, the research community continue to hypothe-
sise and identify new processes and molecular species deemed important to improve our understanding of their impacts. This
continually expanding knowledge base of processes and compounds, however, presents both numerical and computational
challenges on the development of the next generation of mechanistic models. It also raises an important question about appro-
priate design of community driven process models that can not only adapt to increases in complexity, but how we ensure our
platforms exploit emerging computational platforms, if appropriate.

In this paper we present a new community atmospheric 0D box-model, JlBox, written in Julia. Whilst the first version of
JlBox, v1.0, has the same structure and automatic model generation approach as PyBox (Topping et al., 2018), we present
significant improvements in a number of areas. Julia is a relatively new programming language, created with the understanding
that *Scientific computing has traditionally required the highest performance, yet domain experts have largely moved to slower
dynamic languages for daily work.* Julia promises computational performance close to that of Fortran, for example, without
sacrificing flexibility offered by languages such as Python. In JlBox v1.0 we evaluate the performance of a one-language
driven simulation that still utilises automated property predictions provided by UManSysProp and other informatics suites
(Topping et al., 2018). The choice of programming language when building new and sustainable model infrastructures is
clearly influenced by multiple factors. These include issues around training, support and computational performance to name
a few. Python has seen a persistent increase in use across the sciences, in part driven by the large ecosystem and community
driven tools that surrounds it. This was the main factor behind the creation of PyBox. Likewise, in this paper we demonstrate
that the growing ecosystem around Julia offers a number of significant computational and numerical benefits to tackle known
challenges in creating aerosol models using a one language approach. Specifically, we make use of the ability to perform
automatic differentiation of Julia code using tools now available in that ecosystem. In JlBox we demonstrate the usefulness
of this capability when coupling particle phase models to a gas phase model where deriving an analytical jacobian might be
deemed too difficult.

In the following sections we describe the components included within the first version of JlBox, JlBox v1.0. In section
2 we briefly describe the theory on which JlBox is based, including the equations that define implementation of the adjoint
sensitivity studies. In section 3 we discuss the code structure, including parsing algorithms for chemical mechanisms, and the
use of sparse linear solvers and pre-conditioners, whilst also using a range of stiff solvers included within the expanding ODE
55  solver suite DifferentialEquations.jl. In section 4 we then demonstrate the computational performance of JlBox relative to an
existing community gas phase and mixed phase box-models, looking at a range of mechanisms from the Master Chemical
Mechanism (Jenkin et al., 1997, 2002). We present JlBox as a platform for a range of future developments, including the
addition of in/on aerosol processes currently not captured. It is our hope that the demonstration of Julia specific functionality
in this study will facilitate this process.





## 2 Model description

The gas phase reaction of chemicals in atmosphere follows the gas kinetics equation:

$$\frac{d}{dt}[C_i] = -\sum_j r_j S_{ij}, r_j = k_j \prod_{\forall i, S_{ij}>0} [C_i]^{S_{ij}} \tag{1}$$

where $[C_i]$ is the concentration of compound $i$, $r_j$ is the the reaction rate of reaction $j$ , $k_j$ is the corresponding reaction rate coefficient and $S_{ij}$ is the value of the stoichiometry matrix for compound $i$ and $j$. The above Ordinary Differential Equation (ODE), equation (1), fully determines the concentrations of gas phase chemicals at any time given reaction coefficients $k_j$, a stoichiometry matrix $\{S_{ij}\}$ and initial values. All chemical kinetic demonstrations in this study are provided by the Master Chemical Mechanism (MCM) (Jenkin et al., 1997, 2002), but the parsing scheme allows for any mechanism provided in the standard Kinetics PreProcessor [KPP] format (Damian et al., 2002). When adding aerosol particles to the system, more interactions have to be considered in order to predict the state of the system, including concentrations of components in the gas and particulate phase. In JlBox v1.0 we only consider the gas-aerosol partitioning to a fully moving sectional size distribution, recognising the need to use hybrid sectional methods when including coagulation, (e.g., Kokkola et al., 2018). We discuss these future developments in section 5. We use bulk-absorptive partitioning in v1.0 where gas-to-particle partitioning is dictated by gas phase abundance and equilibrium vapour pressures above ideal droplet solutions. This process is described by the growth-diffusion equation provided by Jacobson (2005) (Pages 543, 549, 557, 560).

$$\frac{d[C_{i,k}]}{dt} = 4\pi r_k n_k D_{i,k}^{eff} \left([C_i] - [C_{i,k}^s]\right) \tag{2}$$

$$[C_{i,k}^s] = \exp\left(\frac{2m_{w,i}\sigma}{r_k \rho_i R_* T}\right) \frac{[C_{i,k}]}{[C_{core,k}] \times core\_diss + \sum_i [C_{i,k}]} \frac{p_i^s R_* T}{N_A} \tag{3}$$

$$D_{i,k}^{eff} = \frac{D_i^*}{1 + K_{n,i}\left(\frac{1.33 K_{n,i}+0.71}{K_{n,i}+1} + \frac{4}{3}\frac{1-\alpha_i}{\alpha_i}\right)} \tag{4}$$

where $[C_{i,k}]$ is the concentration of compound $i$ component in size bin $k$, $[C_{i,k}^s]$ is the effective saturation vapor concentration over a curvature surface of size bin $k$ (considering Kelvin effect), $D_{i,k}^{eff}$ is the effective molecular diffusion coefficient, $r_k$ is the size of particles in size bin $k$, $n_k$ the respective number concentration of particles, $[C_{core,k}]$ is the molar concentration of core, $m_{w,i}$ molecular weight of condensate $i$, $\rho_i$ is liquid phase density, $p_i^s$ is pure component saturation vapor pressure, $D_i^*$ is molecular diffusion coefficient, $K_{n,i}$ is Knudsen number, $\alpha_i$ is accommodation coefficient, $\sigma$ is the surface tension of the droplet, $R_*$ is the universal gas constant, $N_A$ is Avogadro's number, and $T$ is temperature.

As we have to keep track of the concentration of every compound in every size bin, this significantly increases the complexity of the ODE relative to the gas phase model:

$$\frac{dy}{dt} = f(y;p), y = (C_1, C_2, \ldots, C_n; C_{1,1}, C_{2,1}, \ldots, C_{n,m}) \tag{5}$$





where $y$ represents the states of the ODE, $n$ is number of chemicals, $m$ is number of size bins, $p$ is a vector of parameters of the ODE, and $f(y)$ is the RHS function implicitly defined by equations (1) and (2). For example, the gas phase simulation of a mechanism with $n = 800$ chemicals has to solve an ODE with 800 states, while the mixed phase simulation with $m = 16$ size

bins will have 13600 ($= 800 + 800 \times 16$) states. Meanwhile, the size of Jacobian matrix (required by implicit ODE solvers) will increase in a quadratic way from $800 \times 800$ to $13600 \times 13600$.

Sensitivity analysis is useful when we need to investigate how the model behaves when we perturb the model parameters and initial values. One approach is to see how all the outputs change due to one perturbed value by simply subtracting the original outputs from the perturbed outputs, or, in a local sense, solving an ODE whose RHS is the partial derivative of the respective

parameter. However, this approach would be very expensive when we want the sensitivity of a scalar output with respect to all the parameters. This is often the case when doing data assimilation. The adjoint method can efficiently solve thie problem. Imagine there is some scalar function $g(y)$ and we would like to compute its sensitivity against some parameters $p$. Introducing the adjoint vector $\lambda(t)$ with the shape of $g(y)$'s gradient, the adjoint method could compute this in two steps (Damian et al., 2002):

1. Solve the ODE (6) in a backward order

2. Numerically integrate formula (7)

$$\frac{d\lambda}{dt} = -\frac{\partial f(y;p)}{\partial y}\lambda, \lambda(t_F) = \frac{\partial g}{\partial y}(t_F) \tag{6}$$

$$\frac{\partial g}{\partial p} = \int_{t_0}^{t_F} \frac{\partial f(y;p)}{\partial p}\lambda(t)dt \tag{7}$$

JlBox implements the adjoint sensitivity algorithm with the help of an auto-generated Jacobian matrix $\partial f(y;p)/\partial y$. Users

only need to supply the gradient function of the scalar function with respect to ODE states $\partial g/\partial y$ as well as the Jacobian function $\partial f(y;p)/\partial p$ of RHS function with respect to parameters so as to get the sensitivity of the scalar function with respect to parameters $\partial g/\partial p$ at time $t_F$. Both are provided automatically through the automatic differentiation provided by Julia.

## 3 Implementation

JlBox written in pure Julia and is presently only dependent on the UManSysProp Python package for parsing chemical struc-

tures into objects for use with fundamental property calculations during a pre-processing stage. The pre-processing stage also includes extracting the rate function, stoichiometry matrix and other parameters from a file that defines the chemical mechanism using the common KPP format, followed by a solution to the self-generated ODEs using implicit ODE solvers. Specifically, the model consists of 6 parts:





1. Run a chemical mechanism parser
2. Perform rate expression formulation and optimization
3. Perform RHS function formulation
4. Create a Jacobian of RHS function
5. Preparation and calculation for partitioning process
6. Adjoint sensitivity analysis where required.

Figure 1 highlights the workflow of an implementation of JlBox, used as either a forward or adjoint model. As detailed in the section on Code Availability, JlBox was designed with both performance and ease of use in mind, where users can download, install and test it as a package from the Julia package manager in the command-line interface. To use the model, one has to construct a configuration object containing all the parameters and initial conditions that the model requires and then supply it to Jl-Box's `run_simulation_*` function. The results are provided as an solution object from `DifferentialEquation.jl`
providing a state vector at any time through interpolation ($\geq 2$ order), along with the respective name vector. Examples can be found in the `example/` subfolder in the project repository which we refer to in section B.

## 3.1 Mechanism parsing and property predictions

**Listing 1.** Example of the MCM Mechanism file

```
{1.}    O = O3 :        5.6D-34*N2*(TEMP/300)**-2.6*O2+6.0D-34*O2*(TEMP/300)**-2.6*O2   ;
{2.}    O + O3 = :    8.0D-12*EXP(-2060/TEMP)          ;
{3.}    O + NO = NO2 :        KMT01   ;
{4.}    O + NO2 = NO :        5.5D-12*EXP(188/TEMP)   ;
{5.}    O + NO2 = NO3 :        KMT02   ;
{6.}    O1D = O :      3.2D-11*EXP(67/TEMP)*O2+2.0D-11*EXP(130/TEMP)*N2         ;
{7.}    NO + O3 = NO2 :        1.4D-12*EXP(-1310/TEMP)          ;
{8.}    NO2 + O3 = NO3 :        1.4D-13*EXP(-2470/TEMP)          ;
{9.}    NO + NO = NO2 + NO2 :  3.3D-39*EXP(530/TEMP)*O2        ;
{10.}   NO + NO3 = NO2 + NO2 :        1.8D-11*EXP(110/TEMP)    ;
```

Like PyBox, JlBox builds the required equations to be solved by reading a chemical mechanism file. In the examples provided here, we use mechanisms extracted from the Master Chemical Mechanism [MCM] to build the intended model for
simulation. A preview of a mechanism file is given in listing 1. There are two sections in each line of the mechanism file separated by the : symbol: the first represents a single gas-phase chemical reaction where reactants before the =" symbol will react with each other with a fixed ratio and produce the products after the = symbol. For example, `a A + b B = c C + d D` represents `a` units of A and `b` units of B will react and produce `c` units of C and `d` units of D.

Upon reading each set of equations, JlBox will assign unique numbers for reactants and products if encountered for the first
time; then it will fill in the stoichiometry matrix $S_{ij}$ with stoichiometry coefficients where $i$ is the number of the equation (depicted at the beginning of each line) and $j$ is the number of the reactants/products. The stoichiometry matrix is firstly built as a list of triplet $(i, j, S_{ij})$ for fast insertion of elements and then it is transformed into the compressed sparse column (CSC) format which is more memory efficient for calculating the RHS of gas-kinetics.





The latter part of a line of the chemical mechanism file, after the symbol `:`, represents the expression of reaction rate coef-

ficient $k_j(y; p)$. The expression consists of prescribed combinations of basic arithmetic operators `+ - * / **`, basic math functions, photolysis coefficients `J(1) ... J(61)`, ambient parameter and intermediate variables which have explicit expressions determined by the chemical mechanism. The drawback of this approach is that pre-processing is separated from simulation, and automatic code generation could, in theory, introduce errors that are hard to debug. However, such a drawback is avoided in JlBox with the help of Julia's meta-programming which assembles the function for calculating reaction rate coef-

ficients 'on the fly'. Since the abstract representation of the function is in the tree format, JlBox also does constant folding optimization to the function where expressions are replaced by their evaluated values if all of the values inside the expressions are found to be constants. For example, the expression `1.2*EXP(1000/TEMP)` will be replaced by `34.340931863060504` given a constant temperature at 298.15K. To further reduce computation, when a reaction rate coefficient is constant, the related expression is deleted from the function which is called at every time step to update the coefficients and the respective initial

value of the coefficient is set to be the constant.

The gas-aerosol partitioning process requires additional pre-processing of several parameters of each compound required by the growth equation. These are listed in equations 2 to 4. Python packages UManSysProp (Topping et al., 2018) and OpenBabel (O'Boyle et al., 2011) are called during the pre-processing stage to calculate thermodynamic properties required by those parameters.

## 3.2 Gas kinetics and gas-aerosol partitioning process

When solving the ODE, the RHS function of the gas phase kinetics firstly updates the non-constant rate coefficients $k_j(y; p)$, then constructs the reaction rate $r_j$ from concentrations of compounds $[C_i]$, their stoichiometry matrix $(S_{ij})$, and rate coefficients $k_j$.

$$r_j = k_j \prod_{\forall i, S_{ij} > 0} [C_i]^{S_{ij}} \tag{8}$$

Following this, the model calculates the rate of change (loss/gain) of reactants and products in each equation and sums the loss/gain of the same species across different equations using:

$$\frac{d}{dt}[C_i] = -\sum_j r_j S_{ij} \tag{9}$$

There are two ways to implement this. The first projects the structure to program instructions executed by the RHS function. The second stores it as data and the RHS function loops through the data to calculate the result.

The first method is intended to statically figure out the symbolic expressions of the loss and gain for each species as combinations of rate coefficients and gas concentrations, and generate the RHS function line by line from the relevant expressions. This method is straightforward and fast, especially for small cases. However, it consumes lots of memory and time for compiling when the mechanism file is large (i.e., > 1000 equations).




The other approach is to use spare matrix manipulation because of the sparse structure of the stoichiometry matrix in
atmospheric chemical mechanisms. Considering equation numbers as columns, compounds numbers as rows, and signed stoichiometry (positive for products and negative for reactants) as values, most columns of the stoichiometry matrix have limited (usually $\leq 4$) nonzero values because most equations have limited number of reactants and products. Therefore, the accumulated rate of change of each compound can be expressed as a sparse matrix-vector product of the stoichiometry matrix and the rates of equations vector while the rates of equations vector can be calculated by loops with cached indices. This method has
comparable speed as the previous one and consumes much less memory when compiling and running.

$$\frac{4}{3}\pi n_k \rho_k r_k^3 = m_k, m_k = m_{core,k} + \sum_i m_{i,k}, m_{i,k} = \frac{m_{w.i}[C_{i,k}]}{N_A} \tag{10}$$

$$\rho_k = \left( \sum_i \frac{m_{i,k}}{m_k \rho_i} + \frac{m_{core,k}}{m_k \rho_{core}} \right)^{-1} \tag{11}$$

The gas-aerosol partitioning component of JlBox simulates the condensational growth of aerosols in discrete size bins where each particle has the same size. Please note that as we use a fully moving distribution in v1.0, when we further refer
to a size bin we retain a discrete representation with no defined limits per bin. JlBox computes the rate of loss/gain for gas phase and condensed phase substances through all size bins. Firstly, for each size bin $k$, the corresponding concentrations of each compound in the condensed phase $\{[C_{i,k}] | \forall i\}$ are summed. Then the model calculates all the values required by the RHS of (2). As we adopted the moving bin scheme in v1.0, it keeps track of the bin sizes $r_k$ as they grow during the process following formulas (10) and (11). Finally, the rate of change of a given specie $d[C_{i,k}]/dt$ is summed across all bins to give the
corresponding loss/gain of gas phase concentrations according to conservation law.

$$\frac{d}{dt}[C_i] = -\sum_j r_j S_{ij} - \sum_k \frac{d}{dt}[C_{i,k}] \tag{12}$$

The combination of the gas phase (9) and condensed phase (12) rate of change expressions provides the overall RHS function (5) of a mixed-phase simulation.

### 3.3  Numerical methods and automatic differentiation

JlBox uses the `DifferentialEquations.jl` library to solve the ODE, assembling the RHS function in a canonical way:
`function dydt!(dydt::Array{<:Real,1}, y::Array{<:Real,1}, p::Dict , t::Real)`. There is a
variety of solvers (>100) available in the `DifferentialEquations.jl` package, from which we generally choose semi-implicit/implicit solvers including Rosenbrock, SDIRK and BDF types of solvers as our problem is numerically stiff. Most of the available solvers are adaptive meaning that they would choose every time step in an adaptive sense to achieve some absolute
and relative errors given by the user. Higher error tolerance allows larger time steps, resulting in faster simulation time and vice





versa. The error tolerance could also influence the convergence of fully implicit ODE solvers due to the non-linear nature of the ODE, so it may fail to converge if the tolerance is too high. Note that native Julia ODE solvers in the `OrdinaryDiffEq.jl` sub-package make use of the parallel (dense) linear solver while the CVODE_BDF solver in `Sundials.jl` sub-package does not. This could mean that the native TRBDF2 solver could be faster than CVODE_BDF on multiprocessor machines,

although they adopt similar algorithms. This would need to be profiled across a range of examples.

Since all the states in the ODE (1, 2) represent the atmospheric abundance of compounds in each phase, it is important to preserve the non-negativeness of those states. This can be ensured by rejecting any states with negative figures and shrinking the time step. Users can specify whether to enable it in the configure object and it is only available in native Julia solvers in the `OrdinarDiffEq.jl` subpackage.

The Jacobian matrix of the RHS $\partial f(y;p)/\partial y$ is needed in implicit ODE solvers as well as in adjoint sensitivity analysis. The accuracy of the Jacobian matrix, however, has variable requirements in each case. For implicit ODE solvers, when doing forward simulations, the accuracy of the matrix only affects the rate of convergence instead of the accuracy of the result. Some methods like BDF and Rosenbrock-W, by-design, could tolerate inaccurate Jacobian matrices (Wanner and Hairer, 1996, p114). Meanwhile, for adjoint sensitivity analysis, accurate Jacobian matrices are needed as they explicitly appear in the RHS function

220 (6).

JlBox implements an analytical Jacobian function for both gas kinetics and partitioning process as well as those generated using finite differentiation and automatic differentiation. Theoretically, an analytical Jacobian is the most accurate and efficient approach, but can be laborious to implement due to the nature of the equations involved and therefore error-prone due to manual imputation. The finite difference approximation can have low numerical accuracy and high performance costs due to multiple

evaluations of the RHS function, although it is the simplest to implement and is applicable to most functions. Automatic differentiation shares the advantages of both methods mentioned previously; it has the convenience of automatically generating a Jacobian matrix from the Julia based model, much like the finite difference method, whilst retaining the accuracy of the analytical solution. Based on the fact that all programs are combination of primitive instructions, an auto-differentiation library could generate the derivative of a program according to the chain rule and predefined derivatives of primitive instructions.

The only limitation is that the RHS function must be fully written in the Julia language and this dictates any additional work that might be required. JlBox uses the `ForwardDiff.jl` library to perform auto-differentiation. The library introduces the dual-number trick with the help of Julia's multiple dispatch mechanism.

To improve performance and reduce memory consumption, JlBox has special treatments for computing the Jacobian of mixed phase RHS. Firstly, the gas kinetic part $\partial f_i/\partial y_j|_{1 \leq i,j \leq n}$ is produced analytically because it is sparse and has simple

analytical form, while auto-differentiation tools will waste lots of memory and time as they treat it as a dense martix. Secondly, according to (12), one part of the Jacobian could be expressed as the sum of another part:

$$\frac{\partial f_i}{\partial y_j}|_{1 \leq i \leq n, n+1 \leq j \leq n+nm}| = \frac{\partial}{\partial y_j} \left( -\sum_{ni+1 \leq k \leq (n+1)i} \frac{d}{dt} y_k \right) = -\sum_{ni+1 \leq k \leq (n+1)i} \frac{\partial f_k}{\partial y_j} \tag{13}$$





which could also reduce computation. We only have to compute the Jacobian of (2) using methods mentioned previously. For comparison of performance and accuracy, JlBox implements two auto-differentiated Jacobians for aerosol processes called

"coarse_seeding" and "fine_seeding" with and without the optimizations mentioned above. According to benchmark results presented in Appendix table B1, it was found those optimizations could significantly reduce memory usage without effecting the performance.

### 3.4 Sparse linear solvers and pre-conditioners

As the size of the Jacobian matrices grow quickly ($O(n^2)$) following the growth of number of states $n$, it becomes increasingly
slow when simulating a mixed-phase model on the full MCM mechanism which has 47544 states when using 16 size bins. The majority of time is spent in solving the dense linear equation $Mx = b$ where $M = I - \gamma J$, $J$ is the Jacobian matrix, $\gamma$ is a scalar set by ODE solver, $x$ and $b$ are some vectors.

Following the Kinetic PreProcessor (KPP) and AtChem model approach (Sommariva et al., 2020; Damian et al., 2002), as the Jacobian is quite sparse JlBox introduces the option to use sparse linear solvers provided by 'DifferentialEquations.jl'.
Specifically JlBox is optimized for the iterative sparse linear solver GMRES in CVODE_BDF by providing pre-conditioners which could drastically reduce the number of iterations of iterative sparse linear solvers like GMRES. Theoretically, a pre-conditioner $P$ is a rough approximation of the matrix $M$ so that $P^{-1}M$ has less condition number than $M$. It is 'rough' in a way that the pre-condition process of solving $P^{-1}x = b$ is easier. In practice, the pre-conditioner $P$ is stored in LU factored form so that solving $P^{-1}x = b$ is a simple back substitution that sometimes needs to be updated to retain proximity with the
changing Jacobian.

In JlBox, the functions for solving $P^{-1}x = b$ and updating $P$ are specified by 'prec' and 'psetup' arguments inside the CVODE_BDF solver. JlBox provides default `prec` and `psetup` as a tri-diagonal pre-conditioner following the approach used in AtChem (Sommariva et al., 2020). In `psetup`, a full Jacobian is calculated in sparse format followed by taking its tridiagonal values forming the approximated tridiagonal $M$. A LU factorization is then calculated using the Thomas algorithm
and stored in cache so that `prec` can solve the linear equation quickly.

### 3.5 Adjoint sensitivity analysis

In this section we simply demonstrate the ability to build and deploy an adjoint model. Using it to quantify sensitivity typically relies on experimental data and processes that will be incorporated in future versions. Nonetheless, the example given in section 4.2 demonstrates the ability to evaluate the sensitivity of predicted secondary organic aerosol to all gas phase kinetic
coefficients. An adjoint sensitivity analysis computes the derivatives of a scalar function $g(y)$ of the ODE states with respect to some parameters $p$ of the RHS function $f(y; p)$. The actual computation reformulates solving the ODE (6) in a backward order and numerically integrating formula (7). It is worth noting that the equation is in the linear form, so using an implicit method that linearizes the RHS function like the Rosenbrock method may give a good result. The Rosenbrock method explicitly includes the Jacobian function as an estimation of the RHS function. In this case, such estimation is an exact representation
which enables longer time-steps. The backward differentiation formula (BDF) may also benefit from this for the same reason,





with the number of Newton steps reduced to one or two. As the Jacobian matrix is frequently called, a fast and accurate Jacobian function is needed. With this in mind, the special treatment of AD mentioned in section 3.3 delivered a 10x improvement in performance compared with the one that simply wrap the RHS with the AD function. For the second step, we adopt the adaptive Gauss-Kronrod quadrature to calculate the formula accurately.

Solving the ODE (6) in a backward manner poses a significant problem as we need to evaluate (an accurate) Jacobian matrix in a backward order (from $t_F$ to $t_0$) which requires accessing the states $y(t)$ at given time-points in backward order. The only way to achieve that is to store a series of states $y_i$ at some checkpoints $t_i$. The stored states alone are sufficient for using ODE solvers with fixed time step, but an adaptive ODE solver is needed for better error control which requires accessing $y(t)$ at an arbitrary time $t$. Thus we need to interpolate those states into dense outputs. Since the time derivative of $y$ is easily accessible

in the form of $dy/dt = f(y)$, we can use Hermite interpolation or higher order interpolation to enhance the accuracy of the interpolation. JlBox utilises the solution object of `DifferentialEquations.jl` (which internally implements Hermite interpolation) to provide $y(t)$ at any given point $t_0 \leq t \leq t_F$.

## 4   Model Output

The goal of JlBox is to provide a high performance mechanistic atmospheric aerosol box model that also retains the flexibility

and usability of Python implementations, for example. Not only should it have comparable performance, if not run faster, than other models for a given scenario, but have the capacity for integrating benchmark chemical mechanisms with coupled aerosol process descriptions. In this section we validate the output of JlBox against PyBox since the model process representations are identical, whilst also investigating the relative performance as the 'size' of the problem scales.

### 4.1   Validation against an existing model box-model

To test the numerical correctness of JlBox, we ran our model together with existing box-model including PyBox and KPP with identical scenarios. JlBox is designed as a more efficient version of PyBox, so it is expected to have identical results in both gas and mixed phase scenarios. Meanwhile, gas phase models constructed from the widely used KPP software could provide some guarantee that the results from JlBox is useful. However, aerosol processes are not available in KPP, as a result we could only compare outputs of gas kinetics. We prepared two test scenarios with gas phase simulation only and mixed-phase simulation.

The settings of the simulations are listed in Table 1. Additionally, in the mixed phase simulation, we set the initial aerosol to be an ideal representation of ammonium sulphate solution satisfying a lognormal size distribution with an average size of 0.2 microns and a standard deviation of 2.2 microns, discretized into 16 bins. The saturation vapour pressure threshold of whether to include the gas-to-particle partitioning of a specific chemical is chosen to be $10^{-6}$ atm based on an extremely low absorptive partitioning coefficient for a wide range of pre-existing mass loadings. For all simulations presented in this paper

we use the vapour pressure technique of Joback and Reid (1987). Whilst known to systematically under predict saturation vapour pressures for species of atmospheric interest (Bilde et al., 2015), we use it for illustrative purposes here and any of the methods included within UManSysProp can be called within JlBox. For gas phase only simulations, we use alpha-pinene as





an indicative VOC degradation scheme. The simulations to compare JlBox with PyBox and KPP are performed on a PC with
a CPU of 8-core AMD Ryzen 1700X at 3.6GHz and 16 Gb RAM.

Figure 2 clearly shows that JlBox and PyBox produced identical results, as designed. Although very close, there is around
1% deviation between the KPP generated model and the other two models. Possible explanation includes differences between
ODE solvers as JlBox & PyBox used CVODE while KPP used LSODEs. For mixed phase simulations, JlBox and PyBox again
generate identical values for secondary organic aerosol mass, as expected.

### 4.2    Evaluation of adjoint sensitivity analysis

A demonstration of an adjoint sensitivity analysis is conducted to calculate the partial derivative of secondary oraganic aerosol
mass (SOA) at the end of simulation with respect to the rate coefficients of each equation in the mechanism. The configurations
of the simulation is the same as the mixed phase alpha-pinene scenario (Table 1) presented in the previous section.

The results presented in Table 2 highlighted the top 10 (in terms of absolute magnitude) estimated deviations of SOA
mass $dSOA$ under a 1% change of rate coefficients because the derivate itself ($dSOA/dratecoeff$) is not comparable due

to different units involved. The reactions between alpha-pinene and ozone have the most substantial effect. The order of the
equations simply highlights the flow of alpha-pinene to its subsequent products. This might attribute to the fact that the system
hasn't reached the equilibrium state (also illustrated in the exponential growth of SOA mass in Figure 2). Another interesting
point is that competing reactions have similar sensitivities but opposite signs like reactions of APINOOB, APINOOA, and
APINENE+OH. The competing reactions between alpha-pinene and ozone is an outlier with a ratio of 5 between the two. A

plausible explanation is that for those reactions with opposite sensitivities, the products of one leads to little or no SOA while
the other contributes more, so when the former reaction is accelerated due to its perturbed rate coefficient, it reduces the ability
of the latter reaction to produce SOA. As a result, the two reactions have opposite sensitivities. For the reactions of APINENE
and O3, it is possible that the APINOOA and APINOOB pathways both produce SOA, and the first produces more than the
second one. When the rate coefficient of the second reaction is increased, the decrease of SOA due to less APINOOA does not

offset the increase of SOA due to more APINOOB, which leads to a smaller but still positive sensitivity of SOA. As we state
earlier, a deeper analysis with alternative options for saturation vapour pressures and process inclusion may reveal important
dependencies.

### 4.3    Performance on large scale problems

In this section we demonstrate the performance of JlBox on 'large scale' problems where both KPP and Pybox fail to solve

due to constraints imposed by the model workflow and language dependencies as shown in Appendix B. We define 'large
scale' problems as those beyond single VOCs or gas phase only simulations. Equipped with a sparse linear solver and auto-
generated tridiagonal preconditioner, JlBox is ideal for simulating larger mechanisms than we present above. With this in
mind, the largest possible mechanism accessible from the MCM suite is selected, which contains 16701 chemical equations,
5832 species. Moreover, we performed 72-hour mixed phase simulations with 16 moving bins. This means that JlBox has to

solve a system of stiff ODEs of 47544 variables that requires solving matrices of $47544 \times 47544$ at each time step. The initial





conditions are taken from an existing representative chamber study on mixed VOC systems (Couvidat et al., 2018, Table 1 & 2) (see Appendix A) with 16 experiments in two sets. We use average values of temperature where ranges are provided. In addition, instead of using the relative humidity selected in those studies, we performed perturbed simulations with low RH scenario of 10% and high RH scenario of 80% respectively to investigate possible dependence on stiffness according to
variable partitioning from the gas to the condensed phase. All the simulations were executed on the ETH Zurich Euler cluster, requesting 4 cores and 7GB memory each to exploit parallelism between different initial conditions. This was chosen as a PC would have to run them in sequential order making it too time consuming.

The elapsed time taken by JlBox is plotted in Figure 3. The average time is around 7 hours which is approximately 1/10 of simulation time. In addition, the maximum memory consumption is 8216 MB and average consumption is 4273 MB. Both
values are significantly smaller than the Jacobian matrix if we were to store it in a double precision dense form. Note that the Euler cluser provide 3 types of CPU nodes equipped with Intel XeonE3 1585Lv5, XeonGold 6150 and AMD EPYC 7742 and the simulation jobs are distributed to all three kinds of node. Although XeonE3 has better single core performance compared to the other two, the time variations between different scenarios far exceeds the variations due to the difference in processors.

Figure 4 shows the generation of SOA mass in the 72-hour period. JlBox captures a diurnal change of photolysis rate as
is depicted in experiment A. We remind the reader that we have no despositional loss, or variable emissions, and that we are using the boiling point method of Joback and Reid (1987) for estimating saturation vapour pressures.

## 5 Discussion

### 5.1 Comparison with other models

JlBox is developed based on the PyBox model (Topping et al., 2018): they have similar structures, rely on the same methods
for calculating pure component properties and provide almost identical results. Despite these similarities, we feel JlBox has made significant improvements over PyBox in terms of readability, functionality, scale-ability, and efficiency from both a programming and algorithmic sense (Table 3). The Julia programming language makes the most significant contribution to those improvements in that it promises a high performance environment, close to Fortran, without sacrificing flexibility of Python. For example, the directly translated partitioning code in JlBox can run at a comparable speed as the individual Fortran
routines in PyBox, and the multiple dispatching mechanism makes it trivial for implementing the automatic differentiation. As a result, JlBox elegantly solves the "two-language problem" without compromising anything by writing everything in Julia. It spares users from editing "code in code" like PyBox that makes it easier to maintain the code base and to extend the model. The homogeneous code base of JlBox also enables a convenient migration to other devices like GPUs considering there is already a GPU backend for Julia.

As for algorithmic advances, the automatic differentiation method for generating Jacobian matrices is not only the most effective addition but also a fundamental one. It is an accurate and convenient way to calculate the Jacobian matrix which only requires an RHS function fully written in Julia. With Jacobian matrices available, the number of RHS evaluations is dramatically reduced since the implicit ODE solver no longer needs to estimate the Jacobian matrix using finite differences.





Also, without automatic differentiation, it will not be so easy to build the adjoint model of a fully coupled process model which
explicitly requires the Jacobian matrix for the entire model, let alone to extend the model with more processes. Besides, the
adaptation of sparse matrices for gas kinetics reduced the compilation cost to a small constant value enabling the JlBox to
simulate large scale mechanisms such as the entire MCM mechanism, which for PyBox typically remains limited by memory.

Compared to other models like KPP (Damian et al., 2002) and AtChem (Sommariva et al., 2020), JlBox is unique due to its
ability to perform coupled mixed phase simulation efficiently especially on large mechanisms such as the full MCM mechanism
where the vanilla KPP variant often fails to compile. JlBox is written in pure standard Julia without any string manipulation to
codes as against KPP and AtChem, which enables full IDE support making it more developer friendly.

## 5.2 Future development

There are a number of processes and algorithmic implementations not included in this version of JlBox that would be useful for
further use in a scientific capacity. These include coagulation, hybrid sectional methods and auto-oxidation products schemes
to name a few (Ehn et al., 2014; Hallquist et al., 2009; Riemer et al., 2009). As we state earlier, the purpose of this develop-
ment stage was to create and profile the first Julia implementation of an aerosol box-model for the scientific community that
would demonstrably exploit the exciting potential this emerging language has to offer. In version 1.0 we provide a fully cou-
pled model. We could, and will, provide options for implementing simplified approaches to aerosol process, such as operator
splitting. Indeed, these methods have proven to provide robust mechanisms for mitigating computational efficiency barriers if
implemented correctly. However our ethos with JlBox is to build and develop a platform for a benchmark community box-
model that exploits the benefits that aforementioned benefits that Julia provides. This includes the ability to exploit existing
and emerging hardware and software platforms as we try to tackle the growing chemical and process complexity associated
with aerosol evolution. We hope that, with version 1.0, the community can help develop and expand this new framework.

Quantifying the importance, or not, of process and chemical complexity requires a multifaceted approach. With the prolif-
eration of data science driven approaches across most scientific domains, Reichstein et al. (2019) note that the next generation
of earth system models are likely going to merge machine learning and traditional process driven models to attempt to solve
aforementioned challenges in complexity whilst exploiting the rich and growing data-sets of global observations. Julia is being
used in development of machine learning (ML) frameworks, with libraries such as Flux-ML enabling researchers to embed
process driven models within the back propagation pipeline (Innes, 2018). This opens up the possibility to develop observa-
tion driven parameterisations in hybrid mechanistic-ML frameworks, which helps with the issue around provenance in ML
parameterisation developments.

JlBox will continually grow and we encourage uptake and further developments.

## Appendix A: Initial condition of section 4.3






**Table A1.** Initial condition for biogenic VOC experiments from Couvidat et al. (2018). Concentrations in ppb, temperature (T) in Kelvin and relative humidity in %

| Experiment | Isoprene | $\alpha$-Pinene | Limonene | NO | NO$_2$ | HONO | SO$_2$ | T | RH |
|---|---|---|---|---|---|---|---|---|---|
| B1 | 107 | 66 | 58 | 34 | 128 | 99 | 0 | 302–307 | 0.5–3 |
| B2 | 92 | 50 | 50 | 48 | 0 | 87 | 0 | 298–300 | 30–26 |
| B3 | 122 | 71 | 40 | 41 | 0 | 53 | 0 | 297–300 | 19–22 |
| B4 | 0 | 63 | 65 | 32 | 0 | 101 | 0 | 294–298 | 8–13 |
| B5 | 99 | 59 | 53 | 150 | 0 | 307 | 0 | 295–297 | 8–11 |
| B6 | 87 | 50 | 51 | 244 | 89 | 40 | 513 | 295–300 | 15–19 |
| B7 | 55 | 79 | 76 | 198 | 0 | 165 | 461 | 302–305 | 20–30 |

**Table A2.** Initial condition for anthropogenic VOC experiments from Couvidat et al. (2018). Concentrations in ppb, temperature (T) in Kelvin and relative humidity in %

| Experiment | Toluene | o-Xylene | TMB | Octane | NO | NO$_2$ | HONO | T | RH |
|---|---|---|---|---|---|---|---|---|---|
| A1 | 102 | 22 | 153 | 85 | 19 | 0 | 99 | 299–305 | 10–16 |
| A2 | 200 | 49 | 300 | 155 | 23 | 0 | 75 | 302–305 | 9–18 |
| A3 | 48 | 11 | 106 | 42 | 23 | 0 | 71 | 302–307 | 6–14 |
| A4 | 98 | 24 | 160 | 79 | 37 | 0 | 156 | 297–307 | 6–13 |
| A5 | 97 | 21 | 146 | 81 | 4 | 8 | 52 | 297–308 | 7–14 |
| A6 | 93 | 22 | 146 | 78 | 21 | 0 | 94 | 300–308 | 0.4 |
| A7 | 107 | 26 | 160 | 89 | 21 | 0 | 89 | 306–309 | 7–10 |
| A8 | 116 | 29 | 19 | 10 | 57 | 0 | 119 | 302–305 | 15–18 |
| A9 | 81 | 21 | 118 | 65 | 31 | 0 | 90 | 299–303 | 28–37 |

## Appendix B: Performance benchmarking

In table B1, we measured the elapsed time and total allocated memory of simulations using varying ODE solvers and techniques of computing the Jacobian matrix mentioned in section 3.3. We chose two ODE solvers: CVODE_BDF and TRBDF2. CVODE_BDF is part of the Sundials suite developed by Lawrence Livermore National Laboratory. It is a widely used high performance ODE solver suitable for large scale stiff ODE problems. TRBDF2 is a Julia-native library implemented in `OrdinaryDiffEq.jl`. It uses the classical TRBDF2 scheme (Hosea and Shampine, 1996) while benefiting from a high-performance linear solver provided by the Julia community.

In table B2 the elapsed time of Pybox, JlBox and KPP are measured, with initial conditions and parameters in section 4.1 and 4.2. We fine tuned JlBox on its ODE solver options to achieve the best performance. For the APINENE mechanism, CVODE was found to be the fastest on gas phase only simulations, while the Julia-native TRBDF2 solver runs better on both forward



**Table B1.** Elapsed time and total allocated memory of the mixed-phase APINENE simulation in section 4.1 with different ODE solvers and Jacobian matrix evaluation techniques

| | Elapsed time (seconds)/total allocated memory | |
|---|---|---|
| Jacobian type | TRBDF2 | CVODE |
| fine seeding | 38.8/2.82GB | 340/1.30GB |
| coarse seeding | 40.3/8.62GB | 350/14.8GB |
| fine analytical | 35.8/2.66GB | 390/1.43GB |
| coarse analytical | 40.5/2.58GB | 357/721MB |
| finite difference | 48.4/13.1GB | 393/25.5GB |

mixed phase simulation and its adjoint sensitivity analysis. For the full MCM mechanism, due to memory restrictions, the only practical option is to use the CVODE ODE solver with the FGMRES sparse linear solver.

**Table B2.** Performance comparison of Pybox, JlBox and KPP based on elapsed time of forward and adjoint simulation in section 4.1 and 4.2 and simulation of full MCM with the same initial condition

| | Elapsed time (seconds) | | | |
|---|---|---|---|---|
| Mechanism, simulation type | Pybox | JlBox | JlBox adjoint | KPP |
| APINENE, gas only | 2.0 | 4.5 | N/A | 0.5 |
| APINENE, mixed phase | 1065 | 55 | 89 | N/A |
| full MCM, gas only | out of memory | 286 | N/A | fail to compile |
| full MCM, mixed phase | out of memory | 1829 | out of memory | N/A |

*Code availability.* The exact code for JlBox v1.0 used in this paper can be found on Zenodo at: https://doi.org/10.5281/zenodo.4075634. The generated KPP Alpha Pinene model can be found at: https://doi.org/10.5281/zenodo.4075632. The JlBox project GitHub page can

be found at: https://github.com/huanglangwen/JlBox. We also provide scripts for building Docker containers to build and run the exact versions of PyBox [v1.0.1], KPP [v2.1] and JlBox [v1.0] to reproduce results provided in this paper. This includes the use of UmanSysProp [v1.01] and OpenBabel [v2.4.1]. Those scripts can be found at: https://github.com/huanglangwen/reproduce_model, with instructions on how replicate the simulations conducted in this paper. The full specification of dependencies of JlBox used in this paper can be found in jlbox\manifest_details.txt in that resspository. An archived copy of the same repository and information can be found on Zenodo at:

https://doi.org/10.5281/zenodo.4134776. JlBox is open source model, licensed under a GPL v3.0. It is compatible with Julia $\geq$ 1.4, Sundials.jl $\geq$ 4.2.5 and OrdinaryDiffEq.jl $\geq$ 5.36.0. As noted on the project GitHub page, JlBox can also be installed through the Julia package manager which deals with all required dependencies. The PyBox project page can be found at: https://github.com/loftytopping/PyBox. PyBox is an open source model, licensed under GPL v3.0. The KPP project page can be found at: http://people.cs.vt.edu/asandu/Software/Kpp/. KPP is



an open source project, licensed under GPL v2.0. The UmanSysProp project page can be found at:

https://github.com/loftytopping/UmanSysProp_public. UManSysProp is an open source project, licensed under GPL v3.0.

*Author contributions.* JlBox was written, and evaluated, by Langwen Huang. David Topping provided the PyBox model and helped understand the effective design and sustainability of JlBox.

*Acknowledgements.* This work was supported by the EPSRC UKCRIC Manchester Urban Observatory (University of Manchester) (grant number: EP/P016782/1). The authors would like to acknowledge the assistance given by Research IT at the University of Manchester. The

authors would also like to acknowledge the ETH Zurich Euler cluster for supporting large scale simulations.





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

**Table 1.** Description of Initial Conditions

| Mechanism | Alpha-Pinene subset of MCM | |
|---|---|---|
| Initial Condition | 18ppm Ozone, 30ppm Alpha-Pinene | |
| Start Time | 12:00 (noon) | |
| Temperature | 288.15K | |
| Simulation | Gas phase only | Mixed phase |
| Relative Humidity | Ignored | 50% |
| Simulation Period | 10800s | 3600s |
| #States | 305 | $2801(305 + 16 \times 156)$ |



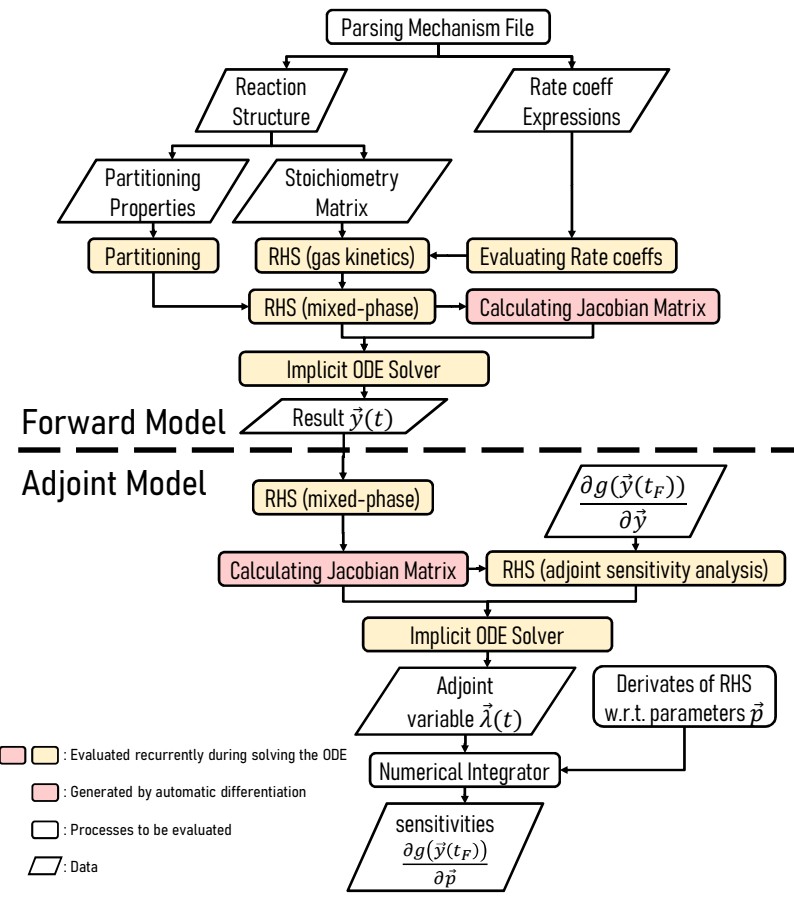

**Figure 1.** Schematic illustrating the structure of JlBoxv1.0, whether in forward or adjoint configuration

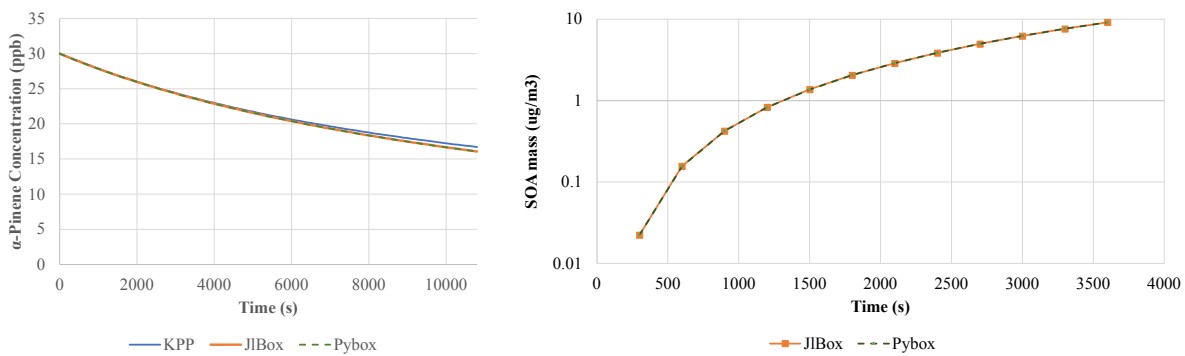

**Figure 2.** Comparison of Gas-only (left) and mixed-phase (right) simulation



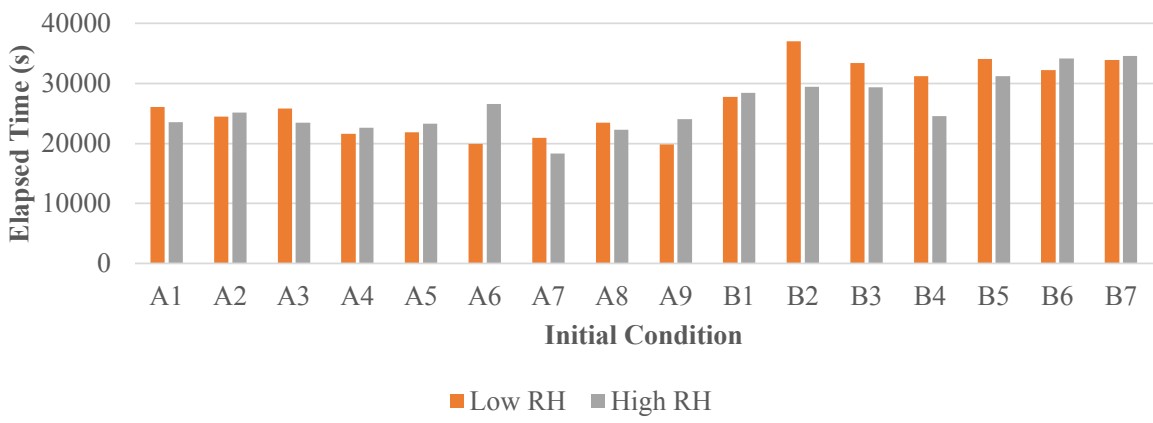

**Figure 3.** Elapsed time of 72h mixed phase simulations. The initial conditions used for each case are listed in the appendix

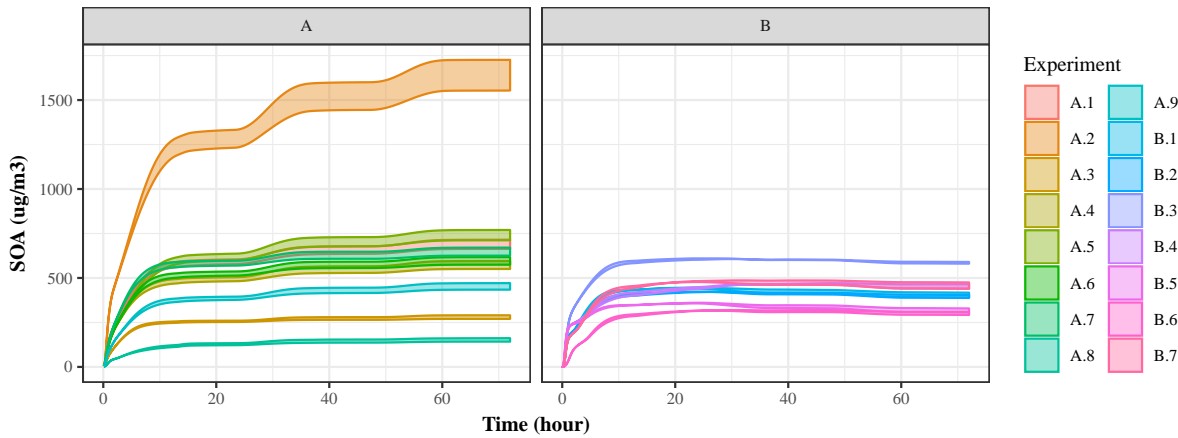

**Figure 4.** Time series plot of SOA mass from the same case studies used in profiling total simulation time. In this study, as noted in the text, we use a predictive technique that under-predicts the saturation vapour pressure to create the maximum number of viable condensing products.





**Table 2.** Sensitivities of SOA mass with respect to gas phase rate coefficients. The units of the last two columns depend on the number of reactants

| Reaction | dSOA ($\mu g/m^3$) | dSOA/dratecoeff | Rate coeff |
|----------|-------------------|-----------------|------------|
| APINENE + O3 = APINOOA | 0.157379287 | 3.003E+17 | 5.240E-17 |
| APINENE + O3 = APINOOB | 0.032264464 | 9.236E+16 | 3.493E-17 |
| APINOOB = C96O2 + OH + CO | 0.006269052 | 1.254E-06 | 5.000E+05 |
| APINOOB = APINBOO | -0.00626905 | -1.254E-06 | 5.000E+05 |
| APINOOA = C109O2 + OH | 0.005857979 | 1.302E-06 | 4.500E+05 |
| APINOOA = C107O2 + OH | -0.00585798 | -1.065E-06 | 5.500E+05 |
| C107O2 = C107OH | 0.005301915 | -1.257E+02 | 4.218E-03 |
| APINENE + OH = APINBO2 | 0.005155068 | 2.643E+10 | 1.950E-11 |
| APINAO2 = APINBOH | 0.005082219 | 8.207E+02 | 6.192E-04 |
| APINENE + OH = APINCO2 | -0.00460746 | -1.112E+11 | 4.144E-12 |



**Table 3.** Comparison between JlBox and PyBox

|  | PyBox | JlBox | Advantage of JlBox |
|---|---|---|---|
| Language | Python+numba or Fortran | Pure Julia | Less code, easier to maintain and extend |
| Parallelization | OpenMP | Parallel Linear Solver | N/A |
| Code generation | Printing string | Meta-programming: generating the abstract syntax tree (AST) | Free syntax check, less human error, easier to maintain |
| Gas kinetics | Static code generation | Sparse matrix manipulation | Much less compiling time, much less memory consumption |
| Property calculation | Python code calling UManSysprop | Translated Julia code calling Umansysprop (python library) | N/A |
| Partitioning | Fortran code | Translated Julia code | Simpler automatic differentiation |
| RHS function | Python code calling Fortran/numba | Julia code | Faster, less memory consumption |
| ODE solver | CVODE_BDF | CVODE_BDF or native solvers | More selections & faster |
| Sparse Jacobian | N/A | Support with GMRES linear solver | Enable large scale mixed phase simulation |
| Jacobian matrix | Handwritten Fortran code for gas kinetics | Handwritten/automatic differentiated/finite differentiated Jacobian for gas kinetics and partitioning, Automatic/finite differentiation can be applied to any additional modules | Less human error, much easier to extend the model, faster mixed phase simulation, enabling local sensitivity analysis based on a Jacobian |
| Sensitivity analysis | N/A | Adjoint sensitivity analysis | Adjoint sensitivity analysis |