# Peer review of "JIBox v1.0: A Julia based mixed-phase atmospheric chemistry box-model"

_Geoscientific Model Development, 2020_

## Referee Comment (RC1) · Anonymous Referee #1 · 29 Nov 2020

The paper presents JlBox - a box model solving the gas phase chemical reactions and the gas - aerosol interactions. The presented model is implemented in Julia, a fairly new high performance general purpose programming language that is well suited for computational science applications. JlBox is based on a Python box model PyBox.

The paper compares the PyBox vs JlBox results and performance. It also discusses the use of different solvers and preconditioners available in JlBox model. As shown throughout the paper, the use of Julia language itself, combined with exploiting different packages available in the Julia ecosystem, significantly improve the JlBox performance over PyBox. An especially interesting case is the application of the automatic differentiation algorithm available in one of the Julia packages to easily build the adjoint sensitivity algorithm in JlBox.

[Figure]

The paper is written well and clearly introduces different options available in the model. The ecosystem of scientific packages available in Julia is growing and I think that JlBox model combined with the description provided in this paper would be a useful new tool for the community. I recommend the paper to be published and I leave some minor comments below.

As a side note - I'm an environmental scientist using Julia in my work, but I'm not an atmospheric chemist. I read the paper and wrote my review from a point of view of a new user trying to use the provided code. However, I cannot review the scientific merit of the presented equations or the numerical approaches used to solve them. Nevertheless, I hope that my comments will be somewhat useful.

Questions:

1) The authors mention that sometimes the time-step length has to be reduced in order to preserve the positive sign of the chemical compounds. I'm wondering how often does that happen and how big of a potential bottleneck could this become. Are there other algorithms that could be implemented to deal with this problem that would not require decreasing the time step?

2) I was wondering if the performance discussion could be extended a little. Table B2 provides simulation times for selected 4 modeling cases. As mentioned before, I'm not an atmospheric chemist, but I was wondering if a plot with the number of chemical compounds, reaction pathways or size bins on one axis and elapsed time on another for the different box models would be useful. This would show how the computational cost scales with the problem complexity for both Julia and Python implementations. I was also wondering, for the large scale simulations performed on the cluster would it be possible to show some scaling plots with the increasing number of processors? Additionally, the authors briefly mention the potential to extend the JlBox to GPU accelerators. What would be the expected performance gain over the current simulations. Are there any other models similar to JlBox that are already running on GPUs?

Minor Comments:

- line 38 - Is the text in italics a quote from somewhere? If yes, could you provide the reference? If not, why is it highlighted?

- ~line 56 - Add section 5 into the list of paper sections described here.

- line 109 - JlBox is written

- line 249 - previously different fonts were used for package names such as DifferentialEquations.jl

- line 262 - I would cross out "simply"

- subsection 4.1 - Maybe the header could be "Validation against existing box-models" the repeated model seems off

- Table 3 - Python library instead of python library, UManSysprop instead of Umansysprop, Numba should be capitalized too

Comments after cloning from the GitHub repo

I tried compiling and running the project following instructions from GitHub. I ran into several small problems, but in general I was able to execute the tests and example simulations. I'm listing the problems I had below as feedback, but it does not concern the manuscript itself.

When running the tests I got an error in the Gas Phase Sparse with Matrix-free operator test complaining about UndefVarError: AnalyticalJacVecOperator not defined. I manually updated DiffEqOperators which solved the problem. After that, all the tests passed.

When running one of the example simulations I got some additional complaints about packages not being installed. I manually added them via Julia package manager following what the error messages were suggesting. In general it seems that the build

and test stage did not set up correctly all the dependencies for me but I was able to easily resolve that.

After that I started getting errors about ../data/*.txt files being missing. The way I understood the "Get Started" section, it suggests executing the simulations from the JlBox folder by include("example/Simulation_*.jl"). But the simulations themselves look for the .txt files in ../data/ folder. The correct way for me to execute the simulations without changing the files was to include("Simulation_*.jl") inside the example folder inside Julia REPL. It might be worth it to update the "Get Started" section on GitHub to clarify that. Everything worked otherwise.

As future work it would be great to include some example plotting scripts in Julia within the GitHub repo and to add a binder setup. This would allow the new users to run and plot the model from the web browser and showcase even better the strength of using Julia where both the high-performance computing and the analysis can be done in one programming language. It would also help a lot to get the new users up to speed in running and visualizing the simulations on their own.

I also think that package naming conventions in Julia suggest not having Julia in their name and instead ask for the package name to finish with .jl extension (https://julialang.github.io/Pkg.jl/v1/creating-packages/#Package-naming-guidelines). Might be too late to suggest changing the package name now, but I thought I should leave it as a comment.

---

## Referee Comment (RC2) · Anonymous Referee #2 · 2 Jan 2021

This paper describes the 0D box-model JLBox, an atmospheric chemistry model to simulate gas phase kinetics and gas-particle partitioning. While plenty such models exist in the community, the innovation of this work is that it is written entirely in Julia geared towards high-performance computing. The paper presents version v1.0 of this model and highlights the advances compared to the python-based model PyBox, which serves as the basis for the JlBox model.

This is interesting work with a lot of potential for future atmospheric chemistry modeling endeavors. The paper is well written and fits well within the scope of GMD. I recommend publication and have some minor suggestions for revisions, mainly to improve the accessibility of this paper for a wider audience.

1. One of the biggest bottlenecks in updating gas/aerosol mechanisms is the interfac-

ing of the chemistry modules with the representation of aerosol microphysics (i.e, bins or modes, what exact bin structure, how many modes etc.) It would be helpful to have some more description in the paper how this is realized within JlBox.

2. Title and throughout the manuscript: The phrase "mixed-phase" to me sounds like referring to a cloud physics model ("mixed-phase clouds"). I suggest using the term "multi-phase" for referring to the combined gas-aerosol system.

3. Eq. (2)-(4): Typesetting of equations: Note that only the subscripts and superscripts that refer to variables should be in italics. Description subscripts and superscripts (eff, w, core, etc.) should be in textmode.

4. Eq. (3): Define variable `core_diss`, and what is considered as "core".

5. Line 80: Rather than "size" of particles, this should read "radius". This applies throughout the manuscript.

6. Line 82: surface tension: Is this the same for all size bins (i.e. taken to be the surface tension of water)?

7. Equation (10): It would be helpful to add some description around this equation and to split up the two equations into two lines. Also switch the lhs and the rhs of the first equation (so that $m_k$ is on the lhs).

8. Line 289: "Validation" should be changed to "Verification", since it refers to the benchmarking with another model.

9. Line 296/297: "average size of 0.2 microns" – I assume the authors mean "geometric mean diameter".

10. Line 297: "microns" should be $\mu$m 11. Line 297: "standard deviation of 2.2 microns" – I assume that this should be the geometric standard deviation. If so, it has the unit 1 (not microns).

12. Line 297: Are the 16 bins logarithmically spaced? And over what radius size

range?

13. Line 317: "exponential growth of SOA mass": The growth doesn't look exponential.

14. Line 339: Regarding the high and low RH scenarios: How is water uptake simulated? Is water one of the n chemicals mentioned in line 86? And what is the reason for different run times depending on RH? Also, the system of equations (2)-(4) assume droplet solutions. Do you assume that the particles always contain water, even at the low RH of 10

15. Line 345: Something went wrong with this sentence, please rephrase.

16. Line 346: should read "cluster provides"

17. Figures 2 and 3: Suggest reporting the time in hours rather than in seconds.

18. Table 2: Suggest reducing the number of sig figs in the dSOA column and in the two last columns write the numbers as scientific notation $3.0 \times 10^{17}$ etc.

19. Even though the simulation results are not of scientific interest in this paper, I suggest including a size distribution plot of the aerosol that undergoes condensational growth for at least the case shown in Figure 2.

20. Switch the order Tables A1 and A2 to make it consistent with the presentation of Figure 3 and 4.

---

## Author Comment (AC1) · 6 Feb 2021

**Response to Anonymous Referee 1**

Langwen Huang and David Topping

Dear colleague. Thank you for taking the time to provide a review of our submitted manuscript, submitted on 29 November 2020. We are pleased your review is supportive of our work and we are of course happy to response to points raised and revise the manuscript accordingly. Please find our responses with any suggested changes in the manuscript below.

*General comment 1): The authors mention that sometimes the time-step length has to be reduced in order to preserve the positive sign of the chemical compounds. I'm wondering how often does that happen and how big of a potential bottleneck could this become. Are there other algorithms that could be implemented to deal with this problem that would not require decreasing the time step?*

**Response:** Negative values occurs when a inaccurate Jacobian is evaluated in an implicit ODE solver at a state with 0 con-
centrations but nonzero tendencies. In practice, this never happens unless the Jacobian or its preconditioner is incorrect or inaccurate. As a result, JlBox almost never shortens the time-step to conserve positivity. However, implicit ODE solvers do decrease the time steps when strong stiffness is detected (e.g. Newton step takes too many iterations, or predicted error is greater than expected). This occurs in some initial conditions. We believe that it is the nature of the ODE system and one cannot do much better compared to current state.

*General comment 2): I was wondering if the performance discussion could be extended a little. Table B2 provides simulation times for selected 4 modeling cases. As mentioned before, I'm not an atmospheric chemist, but I was wondering if a plot with the number of chemical compounds, reaction pathways or size bins on one axis and elapsed time on another for the different box models would be useful. This would show how the computational cost scales with the problem complexity for both Julia and Python implementations. I was also wondering, for the large scale simulations performed on the cluster would it be possible*
*to show some scaling plots with the increasing number of processors? Additionally, the authors briefly mention the potential to extend the JlBox to GPU accelerators. What would be the expected performance gain over the current simulations. Are there any other models similar to JlBox that are already running on GPUs?*

**Response:** Regarding the first point, we also agree it is a good idea to extend the number of cases used to demonstrate performance scaling. In the new manuscript we have now included additional use cases in Figure B1 that plots the simulation
time as a function of number of size bins with different initial conditions (parent VOC and single/mixed option). Whilst each VOC will have varying impacts on the time-to-solution by virtue of the stiffness of the problem and parameters that dictate partitioning, the plot demonstrates that the running time of JlBox with a sparse Jacobian scales roughly linearly with number of size bins. This enables JlBox to perform simulations with higher complexity than was ever possible using PyBox.

Regarding the second point, currently JlBox does not explicitly use multiple cores. While the ODE solver TRBDF2 and
CVODE_BDF do utilise multiple cores, the parallelised component is not a dominant contributor to the total simulation time, making the scaling test easily hitting upper bound of Amdahl's law. In practice, comparative speedups of 200% were only observed in simulations based on the full MCM mechanism that utilised a sparse Jacobian. However, the CPU consumption never exceed 400% in those scenarios. As a result, we did not provide a separate investigation into multi-core use at this stage and allocated 4 cores in all benchmarks.

For the third point, the advantage of running JlBox on a GPU is that the GPU has higher memory bandwidth allowing faster sparse matrix operations. It also has higher floating point computation throughput which is beneficial when inverting dense Jacobian matrices and computing right-hand-side functions. There are efforts to port atmospheric models into GPUs like Linford et al. (2010); Sun et al. (2018); Alvanos and Christoudias (2017). However, these efforts only focus on gas-phase kinetics and small mechanisms designed specifically for use in global chemistry models. As far as we know, there is no other models similar to JlBox that focus on multi-phase and large mechanisms. For sure we agree this is a very interesting area and we should further explore opportunities for parallelisation in the future.

Minor comments

*Minor comment 1): line 38 - Is the text in italics a quote from somewhere? If yes, could you provide thereference? If not, why*

*is it highlighted?*

   **Response:** Yes, our apologies. This text is taken from the Julia documentation. In the new manuscript we will add the following reference: (Julia Documentation: https://julia-doc.readthedocs.io/en/latest/manual/introduction/)

*Minor comment 2): ∼line 56 - Add section 5 into the list of paper sections described here.*

**Response:** Thank you for identifying this. We have now added the following text: In section 5 we discuss the relative merits of JlBox in comparison with other models whilst presenting a narrative on required future developments.

*Minor comment 3): line 109 - JlBox is written*

   **Response:** We have made sure all instances of JlBox are now consistent.

*Minor comment 4): line 249 - previously different fonts were used for package names such as Differen-tialEquations.j*

   **Response:** Apologies, we have now changed the formatting to `DifferentialEquations.jl` to be consistent with other references to Julia packages.

*Minor comment 5): - line 262 - I would cross out "simply*

   **Response:** This has now been deleted.

*Minor comment 6): - subsection 4.1 - Maybe the header could be "Validation against existing box-models" the repeated model seems off*

**Response:** This has now been changed.

*Minor comment 7): - Table 3 - Python library instead of python library, UManSysprop instead of Umansysprop, Numba should be capitalized too*

**Response:** These have now been corrected.

*Comments after cloning from the GitHub repo: I tried compiling and running the project following instructions from GitHub. I ran into several small problems, but in general I was able to execute the tests and example simulations. I'm listing the problems I had below as feedback, but it does not concern the manuscript itself.When running the tests I got an error in the Gas Phase Sparse with Matrix-free operator test complaining about UndefVarError: AnalyticalJacVecOperator not defined. I*

*manually updated DiffEqOperators which solved the problem. After that, all the tests passed.When running one of the example simulations I got some additional complaints about packages not being installed. I manually added them via Julia package manager following what the error messages were suggesting. In general it seems that the build and test stage did not set up correctly all the dependencies for me but I was able to easily resolve that.After that I started getting errors about ../data/\*.txt files being missing. The way I understood the "Get Started" section, it suggests executing the simulations from the JlBox folder*

*by include("example/Simulation_\*.jl"). But the simulations themselves look for the .txt files in ../data/ folder. The correct way for me to execute the simulations without changing the files was to include("Simulation_\*.jl") inside the example folder inside Julia REPL. It might be worth it to update the "Get Started" section on GitHub to clarify that. Everything worked otherwise. As future work it would be great to include some example plotting scripts in Julia within the GitHub repo and to add a binder setup. This would allow the new users to run and plot the model from the web browser and showcase even better the strength*

*of using Julia where both the high-performance computing and the analysis can be done in one programming language. It would also help a lot to get the new users up to speed in running and visualizing the simulations on their own. I also think that package naming conventions in Julia suggest not having Julia in their name and instead ask for the package name to finish with.jlextension (https://julialang.github.io/Pkg.jl/v1/creating-packages/#Package-naming-guidelines). Might be too late to suggest changing the package name now, but I thought I should leave it as a comment*

**Response:** We apologize for the inconvenience of using JlBox following outdated documentation. The documentation has subsequently been updated. All examples have been fixed so that they can find the correct data path independent of the working directory. We also appreciate the idea of using Binder, so we have setup a binder link in the repository as well as providing guidelines to build docker images. We have also updated the Zenodo archive snapshots of the repository as a result.

For the naming of JlBox, it is a bit unfortunate that this name was conceived as a successor of PyBox at summer of 2018

when the naming convention was less clear.

**References**

Alvanos, M. and Christoudias, T.: GPU-accelerated atmospheric chemical kinetics in the ECHAM/MESSy (EMAC) Earth system model (version 2.52), Geoscientific Model Development, 10, 3679–3693, 2017.

Linford, J. C., Michalakes, J., Vachharajani, M., and Sandu, A.: Automatic generation of multicore chemical kernels, IEEE Transactions on Parallel and Distributed Systems, 22, 119–131, 2010.

Sun, J., Fu, J. S., Drake, J. B., Zhu, Q., Haidar, A., Gates, M., Tomov, S., and Dongarra, J.: Computational benefit of GPU optimization for the atmospheric chemistry modeling, Journal of Advances in Modeling Earth Systems, 10, 1952–1969, 2018.

---

## Author Comment (AC2) · 6 Feb 2021

**Response to Anonymous Referee 2**

Langwen Huang and David Topping

Dear colleague. Thank you for taking the time to provide a review of our submitted manuscript, submitted on 2nd January 2021. We are pleased your review is supportive of our work and we are of course happy to response to points raised and revise the manuscript accordingly. Please find our responses with any suggested changes in the manuscript below.

5    *General comment 1): 1. One of the biggest bottlenecks in updating gas/aerosol mechanisms is the interface of the chemistry modules with the representation of aerosol microphysics (i.e, bins or modes, what exact bin structure, how many modes etc.) It would be helpful to have some more description in the paper how this is realized within JlBox*

**Response:** Yes, we agree this is one of the biggest bottlenecks. We hope the automated nature of JlBox at least removes some of the challenges in coupling the gaseous and condensed phases. Whilst we only include a fully moving sectional representation

10   in v1.0, we suggest the following addition to the manuscript might help the reader better understand how this coupling is represented numerically. Please note the figure order in the manuscript reflects the new addition and this figure comes after the definitions of the variables displayed which are already defined in the main text:We extend the original ODE state y with concentrations of each chemicals on each size bins. A simple schematic is provided in Figure 1. Imagine there are $n = 800$ components in the gas phase. In the configuration displayed in figure 2, the first 800 cells hold the concentration of each

15   component in the gas phase. If our simulation has 1 size bin, the proceeding cells hold the concentration of each component in the condensed phase. If our simulation has 2 size bins, the proceeding 800 cells hold the concentration of each component in the second size bin and so on.

[Figure]

**Figure 1.** Array layout for ODE states $y$ in Equation 5

*General comment 2): Title and throughout the manuscript: The phrase "mixed-phase" to me sounds like referring to a cloud*

20   *physics model ("mixed-phase clouds"). I suggest using the term "multi-phase" for referring to the combined gas-aerosol system.*

**Response:** This is a very good point, and we have changed this throughout the document, including the title.

Minor comments

25 *Minor comment 1): Eq. (2)-(4): Typesetting of equations: Note that only the subscripts and superscripts that refer to variables should be in italics. Description subscripts and superscripts (eff,w, core, etc.) should be in textmode*

**Response:**. We have now changed the formatting in equations (2)-(4) and (10) to reflect this.

*Minor comment 2): Eq. (3): Define variable $core\_diss$, and what is considered as "core".*

30 **Response:** Apologies. The modified text in the manuscript now reads as follows: $[C_{\text{core},k}]$ is the molar concentration of an assumed involatile core in v1.0 that may dissociate into $core\_diss$ components. For example, for an ammonium sulphate core, $core\_diss$ is set to 3.0. $m_{\text{w},i}$ is the molecular weight of condensate $i$...

*Minor comment 3): Line 80: Rather than "size" of particles, this should read "radius". This applies throughout the manuscript.*

35 **Response:** We have replaced the instance of 'size' in the manuscript where the context is in relation to the size of the particles. Following the reviewers comment below, this has been changed to 'geometric mean diameter'.

*Minor comment 4): Equation (10): It would be helpful to add some description around this equation and to split up the two equations into two lines. Also switch the lhs and the rhs of the first equation (so that mk is on the lhs*

40 **Response:** We have break Equation (10) into separate lines and added explanations of the equation. Yet we think the lhs and rhs of the first line is appropriate as a fully moving bin scheme has to calculate bin sizes at every time step according to that line.

*Minor comment 5): Line 289: "Validation" should be changed to "Verification", since it refers to the benchmarking with*
45 *another model.*

**Response:** Yes we agree. This has now been changed in the manuscript.

*Minor comment 6): - Line 296/297: "average size of 0.2 microns" – I assume the authors mean "geometric mean diameter".*

**Response:** Yes this is correct and we have specified this in the manuscript.

50

*Minor comment 7): - Line 297: "microns" should beµm11. Line 297: "standard deviation of 2.2 microns"– I assume that this should be the geometric standard deviation. If so, it has the unit 1(not microns).*

**Response:** Yes apologies, this has now been removed.

55 *Minor comment 8): - Line 297: Are the 16 bins logarithmically spaced? And over what radius size range?*

**Response:** Yes, the bins are linearly separated in log space where a fixed volume ratio between bins defines the centre of the bin and bin width. The upper and lower size range and required number of bins define the centre (radius) of each bin accordingly. We have now added this description in section 4.2 as follows: ....discretized into 16 bins. The bins are linearly separated in

log-space where a fixed volume ratio between bins defines the centre of the bin and bin width. The upper and lower size range and required number of bins define the centre (radius) of each bin accordingly.

*Minor comment 9): - Line 317: "exponential growth of SOA mass": The growth doesn't look exponential*

**Response:** Yes, apologies for this lack of clarity. It is indeed sub-exponential growth. We have now removed this, and it should not affect the main point of this paper.

*Minor comment 10): - Line 339: Regarding the high and low RH scenarios: How is water uptake simulated? Is water one of the n chemicals mentioned in line 86? And what is the reason for different run times depending on RH? Also, the system of equations (2)-(4) assume droplet solutions. Do you assume that the particles always contain water, even at the low RH of 10*

**Response:** This is a good point and we suggest some clarification is added to the manuscript. We explicitly simulate the partitioning of water between the gaseous and condensed phase following every other condensate. We appreciate this will, perhaps, significantly reduce the runtime of the box-model. However in this instance we wish to retain the explicit nature of the partitioning process before applying any simplifications such as assuming the mole fraction of water is equivalent to the relative humidity. One future expansion would be to run JlBox in a cloud parcel mode which would require the modification of the droplet growth equation to include latent heat release, but we feel the current architecture provides a good indication of the capability of a Julia based implementation. Likewise, with regards to the second comment on low RH, we assume an ideal solution. Another future development will include the ability to account for non-ideality. However this will also require subsequent treatment of dissociation of inorganic ions and a re-profiling of the subsequent computational cost. We suggest the following text is added to the end of section 3.2:

Please note we explicitly simulate the partitioning of water between the gaseous and condensed phase following every other condensate. We appreciate this may significantly reduce the run-time of the box-model. However, in this instance we wish to retain the explicit nature of the partitioning process before applying any simplifications as we briefly discuss in section 5.2

In section 5.2 'Future Developments' we then suggest the following modification: ...*We could, and will, provide options for implementing simplified approaches to aerosol process, such as operator splitting and assume instantaneous equilibration for water in a range of sub-saturated humid conditions. Indeed, these methods..*

*Minor comment 11): Line 345: Something went wrong with this sentence, please rephrase.*

**Response:** We have now changed this sentence to the following: This represents a significant reduction when compared to the memory required to store a Jacobian matrix in a dense double precision format.

*Minor comment 12): Line 346: should read "cluster provides"*

**Response:** This has been changed.

*Minor comment 13): Figures 2 and 3: Suggest reporting the time in hours rather than in seconds*

**Response:** We agree, this has been changed.

95

*Minor comment 14): Table 2: Suggest reducing the number of sig figs in the dSOA column and in the two last columns write the numbers as scientific notation 3.0×1017 etc*

**Response:** This has now been changed.

100 *Minor comment 15): Even though the simulation results are not of scientific interest in this paper, I suggest including a size distribution plot of the aerosol that undergoes condensational growth for at least the case shown in Figure 2*

**Response:** Thanks for suggestion, we have now added an additional figure of size bin plot for one case in Figure 4.

*Minor comment 16): Switch the order Tables A1 and A2 to make it consistent with the presentation of Figure 3 and 4.*

105 **Response:** This has now been changed.

---

## Author Response (AR2)

**Authors final response**

Langwen Huang and David Topping

Dear Sylwester.

Thank you for taking the time to act as editor of our submitted manuscript. We are delighted you feel the paper is now ready for publication. In the following we summarise our response to your final requested minor corrections and then detail the previous responses to the reviewer comments.

With regards to the final minor corrections:

*1) - suggest more precise wording for: "informatics software" (p1/l11) and "numerical and computational challenges" (p2/l32)*

**Response:** Sure. With regards to the first point, we have now modified this sentence to read as follow:

JlBox is built around chemical mechanism files, using existing informatics software to parse chemical structures and relationships from these files and then provide parameters required for mixed phase simulations.

With regards to the second point we have changed the sentence to read as follows:

With ongoing investments in atmospheric aerosol monitoring technologies, the research community continue to hypothesise and identify new processes and molecular species deemed important to improve our understanding of their impacts. This continually expanding knowledge base of processes and compounds, however, likewise requires us to expand our aerosol modelling frameworks to capture this increased complexity

*2) - Pybox is spelled without capital B in the header of Table B2*

**Response:** This has been corrected.

*3) - "it is compatible with" and ">=" version indication might be too optimistic (e.g., Python 2 vs. Python 3 issue, to name a big one) - better to state that it was developed ensuring/testing compatibility with ...?*

**Response:** Yes very good point, the code section now reads as follows: The exact code for JlBox v1.1 used in this paper can be found on Zenodo at:https://doi.org/10.5281/zenodo.4519192. The generated KPP Alpha Pinene model can be found at: https://doi.org/10.5281/zenodo.4075632. The JlBox project GitHub page can be found at: https://github.com/huanglangwen/JlBox. We also provide scripts for building Docker containers to build and run the exact versions of PyBox [v1.0.1], KPP [v2.1] and JlBox [v1.1] to reproduce results provided in this paper. This includes the use of UmanSysProp [v1.01] and OpenBabel [v2.4.1]. Those scripts can be found at: https://github.com/huanglangwen/reproduce_model, with instructions on how replicate the simulations conducted in this paper. The full specification of dependencies of JlBox used in this paper can be found in jlbox\manifest_details.txt in that respository. An archived copy of the same repository and information can be found on Zenodo at: https://doi.org/10.5281/zenodo.4543713. JlBox is open source model, licensed under a GPL v3.0. It has been developed to ensure compatibility with Julia v1, Sundials.jl, OrdinaryDiffEq.jl where detailed dependency information is available in the reproduction repository. As noted on the project GitHub page, JlBox can also be installed through the Julia package manager which deals with all required dependencies. The PyBox project page can be found at: https://github.com/loftytopping/PyBox. PyBox is an open source model, licensed under GPL v3.0. The KPP project page can be found at: http://people.cs.vt.edu/
asandu/Software/Kpp/. KPP is an open source project, licensed under GPL v2.0. The UmanSysProp project page can be found at:

https://github.com/loftytopping/UmanSysProp_public. UManSysProp is an open source project, licensed under GPL v3.0.

*3)- "DT provided the PyBox model" in Author contributions might sound as if it was not open source - please rephrase*

**Response:** This has been modified to read as follows: JlBox was written, and evaluated, by Langwen Huang. David Topping provided guidance on comparisons with PyBox, including mapping the same structure to JlBox and helped understand the effective design and sustainability of JlBox.

*4)- please unify font sizes in figure axis labels. Fig. 5 reads best, Fig. 7 is hardly readable when printed with two-pages per*
*sheet.*

**Response:** This has been corrected

*5)- The title refers to v1.0, while the zenodo archive includes v1.1 - please update the version number in the title.*

**Response:** This has been corrected.

*6)- Also, let me suggest referencing the 2019 Nature "TOOLBOX" paper on Julia*

**Response:** This has been added.

With regards to responses to reviewers: To briefly summarise, reviewer 1 raised some very useful points around the numerical
mechanics of the model and requested more evidence of performance bench-marking which we have now provided in Appendix section B. They also raised a few hurdles in running the code in their own Julia distribution which we have now fixed. Usefully, we have now setup a binder link in the repository as well as providing guidelines to build docker images. Reviewer 2 raised some very important points around domain bottlenecks which we have clarified in the new manuscript, providing a brief clarification of the structure of our numerical arrays. We have also slightly modified the title to refer to a 'multi-phase' rather
than 'mixed-phase' model to avoid confusion with the cloud physics community.

Many thanks
Langwen and David

**Response to Anonymous Referee 1**

Langwen Huang and David Topping

Dear colleague. Thank you for taking the time to provide a review of our submitted manuscript, submitted on 29 November 2020. We are pleased your review is supportive of our work and we are of course happy to response to points raised and revise the manuscript accordingly. Please find our responses with any suggested changes in the manuscript below.

*General comment 1): The authors mention that sometimes the time-step length has to be reduced in order to preserve the positive sign of the chemical compounds. I'm wondering how often does that happen and how big of a potential bottleneck could this become. Are there other algorithms that could be implemented to deal with this problem that would not require decreasing the time step?*

   **Response:** Negative values occurs when a inaccurate Jacobian is evaluated in an implicit ODE solver at a state with 0 con-
centrations but nonzero tendencies. In practice, this never happens unless the Jacobian or its preconditioner is incorrect or inaccurate. As a result, JlBox almost never shortens the time-step to conserve positivity. However, implicit ODE solvers do decrease the time steps when strong stiffness is detected (e.g. Newton step takes too many iterations, or predicted error is greater than expected). This occurs in some initial conditions. We believe that it is the nature of the ODE system and one cannot do much better compared to current state.

*General comment 2): I was wondering if the performance discussion could be extended a little. Table B2 provides simulation times for selected 4 modeling cases. As mentioned before, I'm not an atmospheric chemist, but I was wondering if a plot with the number of chemical compounds, reaction pathways or size bins on one axis and elapsed time on another for the different box models would be useful. This would show how the computational cost scales with the problem complexity for both Julia and Python implementations. I was also wondering, for the large scale simulations performed on the cluster would it be possible*
*to show some scaling plots with the increasing number of processors? Additionally, the authors briefly mention the potential to extend the JlBox to GPU accelerators. What would be the expected performance gain over the current simulations. Are there any other models similar to JlBox that are already running on GPUs?*

   **Response:** Regarding the first point, we also agree it is a good idea to extend the number of cases used to demonstrate performance scaling. In the new manuscript we have now included additional use cases in Figure B1 that plots the simulation
time as a function of number of size bins with different initial conditions (parent VOC and single/mixed option). Whilst each VOC will have varying impacts on the time-to-solution by virtue of the stiffness of the problem and parameters that dictate partitioning, the plot demonstrates that the running time of JlBox with a sparse Jacobian scales roughly linearly with number of size bins. This enables JlBox to perform simulations with higher complexity than was ever possible using PyBox.

   Regarding the second point, currently JlBox does not explicitly use multiple cores. While the ODE solver TRBDF2 and
CVODE_BDF do utilise multiple cores, the parallelised component is not a dominant contributor to the total simulation time, making the scaling test easily hitting upper bound of Amdahl's law. In practice, comparative speedups of 200% were only observed in simulations based on the full MCM mechanism that utilised a sparse Jacobian. However, the CPU consumption never exceed 400% in those scenarios. As a result, we did not provide a separate investigation into multi-core use at this stage and allocated 4 cores in all benchmarks.

For the third point, the advantage of running JlBox on a GPU is that the GPU has higher memory bandwidth allowing faster sparse matrix operations. It also has higher floating point computation throughput which is beneficial when inverting dense Jacobian matrices and computing right-hand-side functions. There are efforts to port atmospheric models into GPUs like Linford et al. (2010); Sun et al. (2018); Alvanos and Christoudias (2017). However, these efforts only focus on gas-phase kinetics and small mechanisms designed specifically for use in global chemistry models. As far as we know, there is no other models similar to JlBox that focus on multi-phase and large mechanisms. For sure we agree this is a very interesting area and we should further explore opportunities for parallelisation in the future.

Minor comments

*Minor comment 1): line 38 - Is the text in italics a quote from somewhere? If yes, could you provide the reference? If not, why*
*is it highlighted?*

    **Response:** Yes, our apologies. This text is taken from the Julia documentation. In the new manuscript we will add the following reference: (Julia Documentation: https://julia-doc.readthedocs.io/en/latest/manual/introduction/)

*Minor comment 2): ~line 56 - Add section 5 into the list of paper sections described here.*
**Response:** Thank you for identifying this. We have now added the following text: In section 5 we discuss the relative merits of JlBox in comparison with other models whilst presenting a narrative on required future developments.

*Minor comment 3): line 109 - JlBox is written*

    **Response:** We have made sure all instances of JlBox are now consistent.

*Minor comment 4): line 249 - previously different fonts were used for package names such as Differen-tialEquations.j*

    **Response:** Apologies, we have now changed the formatting to `DifferentialEquations.jl` to be consistent with other references to Julia packages.

*Minor comment 5): - line 262 - I would cross out "simply*

    **Response:** This has now been deleted.

*Minor comment 6): - subsection 4.1 - Maybe the header could be "Validation against existing box-models" the repeated model seems off*

**Response:** This has now been changed.

*Minor comment 7): - Table 3 - Python library instead of python library, UManSysprop instead of Umansysprop, Numba should be capitalized too*

**Response:** These have now been corrected.

*Comments after cloning from the GitHub repo: I tried compiling and running the project following instructions from GitHub. I ran into several small problems, but in general I was able to execute the tests and example simulations. I'm listing the problems I had below as feedback, but it does not concern the manuscript itself.When running the tests I got an error in the Gas Phase Sparse with Matrix-free operator test complaining about UndefVarError: AnalyticalJacVecOperator not defined. I*

*manually updated DiffEqOperators which solved the problem. After that, all the tests passed.When running one of the example simulations I got some additional complaints about packages not being installed. I manually added them via Julia package manager following what the error messages were suggesting. In general it seems that the build and test stage did not set up correctly all the dependencies for me but I was able to easily resolve that.After that I started getting errors about ../data/*.txt files being missing. The way I understood the "Get Started" section, it suggests executing the simulations from the JlBox folder*

*by include("example/Simulation_\*.jl"). But the simulations themselves look for the .txt files in ../data/ folder. The correct way for me to execute the simulations without changing the files was to include("Simulation_\*.jl") inside the example folder inside Julia REPL. It might be worth it to update the "Get Started" section on GitHub to clarify that. Everything worked otherwise. As future work it would be great to include some example plotting scripts in Julia within the GitHub repo and to add a binder setup. This would allow the new users to run and plot the model from the web browser and showcase even better the strength*

*of using Julia where both the high-performance computing and the analysis can be done in one programming language. It would also help a lot to get the new users up to speed in running and visualizing the simulations on their own. I also think that package naming conventions in Julia suggest not having Julia in their name and instead ask for the package name to finish with.jlextension (https://julialang.github.io/Pkg.jl/v1/creating-packages/#Package-naming-guidelines). Might be too late to suggest changing the package name now, but I thought I should leave it as a comment*

**Response:** We apologize for the inconvenience of using JlBox following outdated documentation. The documentation has subsequently been updated. All examples have been fixed so that they can find the correct data path independent of the working directory. We also appreciate the idea of using Binder, so we have setup a binder link in the repository as well as providing guidelines to build docker images. We have also updated the Zenodo archive snapshots of the repository as a result.

For the naming of JlBox, it is a bit unfortunate that this name was conceived as a successor of PyBox at summer of 2018

when the naming convention was less clear.

**References**

Alvanos, M. and Christoudias, T.: GPU-accelerated atmospheric chemical kinetics in the ECHAM/MESSy (EMAC) Earth system model (version 2.52), Geoscientific Model Development, 10, 3679–3693, 2017.

Linford, J. C., Michalakes, J., Vachharajani, M., and Sandu, A.: Automatic generation of multicore chemical kernels, IEEE Transactions on Parallel and Distributed Systems, 22, 119–131, 2010.

Sun, J., Fu, J. S., Drake, J. B., Zhu, Q., Haidar, A., Gates, M., Tomov, S., and Dongarra, J.: Computational benefit of GPU optimization for the atmospheric chemistry modeling, Journal of Advances in Modeling Earth Systems, 10, 1952–1969, 2018.

**Response to Anonymous Referee 2**

Langwen Huang and David Topping

Dear colleague. Thank you for taking the time to provide a review of our submitted manuscript, submitted on 2nd January 2021. We are pleased your review is supportive of our work and we are of course happy to response to points raised and revise the manuscript accordingly. Please find our responses with any suggested changes in the manuscript below.

*General comment 1): 1. One of the biggest bottlenecks in updating gas/aerosol mechanisms is the interface of the chemistry modules with the representation of aerosol microphysics (i.e, bins or modes, what exact bin structure, how many modes etc.) It would be helpful to have some more description in the paper how this is realized within JlBox*

**Response:** Yes, we agree this is one of the biggest bottlenecks. We hope the automated nature of JlBox at least removes some of the challenges in coupling the gaseous and condensed phases. Whilst we only include a fully moving sectional representation in v1.0, we suggest the following addition to the manuscript might help the reader better understand how this coupling is represented numerically. Please note the figure order in the manuscript reflects the new addition and this figure comes after the definitions of the variables displayed which are already defined in the main text: We extend the original ODE state y with concentrations of each chemicals on each size bins. A simple schematic is provided in Figure 1. Imagine there are $n = 800$ components in the gas phase. In the configuration displayed in figure 2, the first 800 cells hold the concentration of each component in the gas phase. If our simulation has 1 size bin, the proceeding cells hold the concentration of each component in the condensed phase. If our simulation has 2 size bins, the proceeding 800 cells hold the concentration of each component in the second size bin and so on.

[Figure]

**Figure 1.** Array layout for ODE states $y$ in Equation 5

*General comment 2): Title and throughout the manuscript: The phrase "mixed-phase" to me sounds like referring to a cloud*

*physics model ("mixed-phase clouds"). I suggest using the term "multi-phase" for referring to the combined gas-aerosol system.*

**Response:** This is a very good point, and we have changed this throughout the document, including the title.

Minor comments

*Minor comment 1): Eq. (2)-(4): Typesetting of equations: Note that only the subscripts and superscripts that refer to variables should be in italics. Description subscripts and superscripts (eff,w, core, etc.) should be in textmode*

**Response:**. We have now changed the formatting in equations (2)-(4) and (10) to reflect this.

*Minor comment 2): Eq. (3): Define variable $core\_diss$, and what is considered as "core".*

**Response:** Apologies. The modified text in the manuscript now reads as follows: $[C_{\text{core},k}]$ is the molar concentration of an assumed involatile core in v1.0 that may dissociate into $core\_diss$ components. For example, for an ammonium sulphate core, $core\_diss$ is set to 3.0. $m_{\text{w},i}$ is the molecular weight of condensate $i$...

*Minor comment 3): Line 80: Rather than "size" of particles, this should read "radius". This applies throughout the manuscript.*

**Response:** We have replaced the instance of 'size' in the manuscript where the context is in relation to the size of the particles. Following the reviewers comment below, this has been changed to 'geometric mean diameter'.

*Minor comment 4): Equation (10): It would be helpful to add some description around this equation and to split up the two equations into two lines. Also switch the lhs and the rhs of the first equation (so that mk is on the lhs*

**Response:** We have break Equation (10) into separate lines and added explanations of the equation. Yet we think the lhs and rhs of the first line is appropriate as a fully moving bin scheme has to calculate bin sizes at every time step according to that line.

*Minor comment 5): Line 289: "Validation" should be changed to "Verification", since it refers to the benchmarking with*
*another model.*

**Response:** Yes we agree. This has now been changed in the manuscript.

*Minor comment 6): - Line 296/297: "average size of 0.2 microns" – I assume the authors mean "geometric mean diameter".*
**Response:** Yes this is correct and we have specified this in the manuscript.

*Minor comment 7): - Line 297: "microns" should be$\mu$m11. Line 297: "standard deviation of 2.2 microns"– I assume that this should be the geometric standard deviation. If so, it has the unit 1(not microns).*
**Response:** Yes apologies, this has now been removed.

*Minor comment 8): - Line 297: Are the 16 bins logarithmically spaced? And over what radius size range?*
**Response:** Yes, the bins are linearly separated in log space where a fixed volume ratio between bins defines the centre of the bin and bin width. The upper and lower size range and required number of bins define the centre (radius) of each bin accordingly. We have now added this description in section 4.2 as follows: ....discretized into 16 bins. The bins are linearly separated in log-space where a fixed volume ratio between bins defines the centre of the bin and bin width. The upper and lower size range and required number of bins define the centre (radius) of each bin accordingly.

*Minor comment 9): - Line 317: "exponential growth of SOA mass": The growth doesn't look exponential*

**Response:** Yes, apologies for this lack of clarity. It is indeed sub-exponential growth. We have now removed this, and it should not affect the main point of this paper.

*Minor comment 10): - Line 339: Regarding the high and low RH scenarios: How is water uptake simulated? Is water one of the n chemicals mentioned in line 86? And what is the reason for different run times depending on RH? Also, the system of equations (2)-(4) assume droplet solutions. Do you assume that the particles always contain water, even at the low RH of 10*

**Response:** This is a good point and we suggest some clarification is added to the manuscript. We explicitly simulate the partitioning of water between the gaseous and condensed phase following every other condensate. We appreciate this will, perhaps, significantly reduce the runtime of the box-model. However in this instance we wish to retain the explicit nature of the partitioning process before applying any simplifications such as assuming the mole fraction of water is equivalent to the relative humidity. One future expansion would be to run JlBox in a cloud parcel mode which would require the modification of the droplet growth equation to include latent heat release, but we feel the current architecture provides a good indication of the capability of a Julia based implementation. Likewise, with regards to the second comment on low RH, we assume an ideal solution. Another future development will include the ability to account for non-ideality. However this will also require subsequent treatment of dissociation of inorganic ions and a re-profiling of the subsequent computational cost. We suggest the following text is added to the end of section 3.2:

Please note we explicitly simulate the partitioning of water between the gaseous and condensed phase following every other condensate. We appreciate this may significantly reduce the run-time of the box-model. However, in this instance we wish to retain the explicit nature of the partitioning process before applying any simplifications as we briefly discuss in section 5.2

In section 5.2 'Future Developments' we then suggest the following modification: ...*We could, and will, provide options for implementing simplified approaches to aerosol process, such as operator splitting and assume instantaneous equilibration for water in a range of sub-saturated humid conditions. Indeed, these methods..*

*Minor comment 11): Line 345: Something went wrong with this sentence, please rephrase.*

**Response:** We have now changed this sentence to the following: This represents a significant reduction when compared to the memory required to store a Jacobian matrix in a dense double precision format.

*Minor comment 12): Line 346: should read "cluster provides"*

**Response:** This has been changed.

*Minor comment 13): Figures 2 and 3: Suggest reporting the time in hours rather than in seconds*

**Response:** We agree, this has been changed.

*Minor comment 14): Table 2: Suggest reducing the number of sig figs in the dSOA column and in the two last columns write the numbers as scientific notation $3.0 \times 10^{17}$ etc*

**Response:** This has now been changed.

*Minor comment 15): Even though the simulation results are not of scientific interest in this paper, I suggest including a size distribution plot of the aerosol that undergoes condensational growth for at least the case shown in Figure 2*

**Response:** Thanks for suggestion, we have now added an additional figure of size bin plot for one case in Figure 4.

*Minor comment 16): Switch the order Tables A1 and A2 to make it consistent with the presentation of Figure 3 and 4.*

**Response:** This has now been changed.